# Microstructure and Mechanical Performance of 3D Printed Wood-PLA/PHA Using Fused Deposition Modelling: Effect of Printing Temperature

**DOI:** 10.3390/polym11111778

**Published:** 2019-10-29

**Authors:** Sofiane Guessasma, Sofiane Belhabib, Hedi Nouri

**Affiliations:** 1INRA, UR1268 Biopolymères Interactions Assemblages, F-44300 Nantes, France; 2Laboratoire GEPEA, UMR CNRS 6144, Université - IUT de Nantes, avenue du Professeur Jean Rouxel, 44475 Carquefou Cédex, France; sofiane.belhabib@univ-nantes.fr; 3Laboratoires des SystèmesElectromécaniques (LASEM-ENIS), Université de Sfax, Route Soukra Km3, Sfax BPW3038, Tunisia; hedi.nouri@isams.usf.tn

**Keywords:** fused deposition modelling, wood-PLA/PHA, tensile properties, X-ray micro-tomography, finite element computation, printing temperature

## Abstract

The microstructure and mechanical performance of wood-based filament is investigated in the case of Fused Deposition Modelling (FDM) technique using experimental and numerical approaches. The printing process of wood-PLA/PHA is conducted by varying the printing temperature, typically from 210 °C to 250 °C. The filament temperature during the laying down is measured using infra-red camera to study the thermal cycling. In addition, X-ray micro-tomography is used to evaluate the material arrangement of printed wood-PLA/PHA at different length scales. Tensile experiments are performed to rank the loss in mechanical performance with respect to the filament properties. Finally, finite element computation is considered to predict the tensile behaviour based on the implementation of the real 3D microstructure issued from X-ray micro-tomography. The results show that the wood-based filament is printable over a wide range of temperatures and exhibits a marked heat accumulation tendency at high printing temperatures. However, the limited gain in tensile performance at these temperatures makes 220 °C an optimal choice for printing wood-based filament. The elongation at break of 3D-printed wood-PLA/PHA is remarkably similar to the results observed for the filament. Finite element computation reveals that despite this apparent similarity, the associated deformation mechanisms are different.

## 1. Introduction

Fused Deposition Modelling (FDM) is a 3D printing technique used to process polymeric raw filaments based on digitalised models [1]. The process consists of a driving system allowing the raw polymeric filament to go through a nozzle where the polymer is heated to above its glass transition temperature. The fused state allows the extruded filament to be laid down from the nozzle tip. The motion of the entire nozzle in the plane of building controls the drawing of 2D features. When the nozzle is moved upwards with respect to the building platform, a 3D object is formed as a successive set of layers [2]. The simplicity of the building process in FDM makes it a versatile choice to print different grades of polymers if the thermal behaviour of the laying down process is controlled. One of the most important printing parameters in FDM is thus the printing temperature [3,4]. The appropriate choice of printing temperature is important to guaranteeing a good rendering of the printed features, to reduce the effect of the process-induced defects and to bring the performance loss induced by the defect genesis to an acceptable threshold [5,6]. 

The correlation between the mechanical performance and the printing temperature can be understood based on the analysis of the filament arrangement generated during the laying down process [7]. The material discontinuities created within the plane of construction and in the building direction contribute negatively to the cohesion of the 3D print [8]. The lack of cohesion in turn limits the mechanical load transfer by combining two main mechanisms: the stress concentration that develops around the process-generated porosities, and the filament decohesion that occurs at the interface [8]. A high printing temperature contributes to lowering the effect of process-generated porosities and improving the cohesive structure of the 3D print if cleverly combined with other printing parameters, such as part orientation, layer thickness, printing speed, etc. [9,10]. 

Several studies have reported the positive effect of the printing temperature on the mechanical performance of printed polymers. For instance, Kumar et al. [10] studied the effect of the printing temperature on the tensile properties of a low melting copolymer called ethylene vinyl acetate within the range 100–120 °C. Their study concluded a strong positive correlation between the printing temperature and the tensile properties including the tensile strength and elongation at break. Wittbrodt et al. [11] demonstrated that the mechanical strength of polylactic acid (PLA) has a nearly linear correlation with the printing temperature between 190 °C and 215 °C. Aliheidari et al. [12] measured the fracture performance of 3D printed Acrylonitrile Butadiene Styrene (ABS) specimens and showed that the ranking of the mechanical response continuously improves with the increase of the printing temperature within the range (210–240 °C).

Few results are available on the performance of wood-based filaments, especially with respect to the influence of the process parameters, despite its unique characteristics of texture, smell, and surface finish that can be used in engineering FDM-based parts. The recent review by Mazzanti et al. [13] showed that the use of natural filler is a promising route for the design of environmentally friendly components that exhibit superior performance. The analysis of the literature works in the same review paper demonstrated the lack of impactful research on biofillers combined with PLA or ABS compared to less known polymers such as polyolefins. Kariz et al. [14] formulated wood-based filaments with PLA with different contents of wood particles ranging from 10% to 50%. The raw filament exhibited a decreasing tensile strength when the wood content was increased. Compared to PLA, the 10% wood-based filament demonstrated an improved tensile strength. In addition, dynamic mechanical testing of 3D-printed wood-based material demonstrated a low storage modulus for a high wood content in the filament. Yang considered a wood fibre filler in PLA as a feedstock material in FDM [15]. This author showed that the mechanical properties were negatively affected by printing temperatures larger than 200 °C, while most physical properties, including the density of the printed composite, increased with the increase of the printing temperature in the range 200–230 °C. Le Duigou et al. [16] studied the mechanical performance of a commercial wood-based filament similar to the one used in this work. This study revealed the sensitivity of the mechanical performance to the size of the parts and the orientation of the parts with respect to the building direction. In the present study, the effect of printing temperature on tensile performance and the thermal behaviour of wood-based filament is explored. Both experimental and numerical approaches are considered in order to reveal the correlation between the underlined microstructure and the observed performance.

## 2. Experimental Setup

The feedstock material is a 30% recycled pinewood fibre—70% PLA/PHA biosourced composite purchased from FDBI company (Asnieres, France) under the tradename ColorFabb woodfill. This material comes in the form of a wire cartridge with a typical filament diameter of 1.75 ± 0.05 mm. The documented glass transition for this wood-based filament is 55 °C. The supplier recommendations for the printing temperature are between 195 °C and 220 °C, with a heated printing bed from 50 °C to 60 °C. In addition, the recommended printing speed is in the range of 40–100 mm/s. Differential Scanning Calorimetry (DSC) performed in this study confirms a slightly larger glass temperature of about 57 °C (Figure 1a). These measurements are based on a DSC823e equipment from Mettler Toledo (Columbus, OH, USA). The test run from −40 °C to 300 °C with an increment of 10 °C/min additionally revealed distinct chemical activity between 100 °C and 180 °C due to the presence of water in wood. Tensile testing of the as-received filament was undertaken to obtain a reference for the measurement of the expected loss in mechanical performance of the 3D printed specimens. The testing was conducted on single filaments up to the rupture point using a Zwick Roell universal machine. The displacement rate was fixed at 5 mm/min (Figure 1b).

The FDM process was conducted on a Replicator 2 printer from MakerBot, which was controlled using the MakerBot© Desktop software. An extended list of the printing parameters is given in Table 1. The extruder temperature, referred to as the printing temperature, was varied between 210 °C and 255 °C.

The geometry of the tensile specimens was selected according to the norm ISO 527-1/2. The orientation of the specimens was selected to allow the best tensile performance, as shown in several previous studies [2,17,18]. 

This means that the tensile loading direction was normal to the building direction, and the printing was conducted along the smallest dimension (sample thickness aligned with the building direction). In the literature, this is referred to as horizontal orientation [2]. Four samples were printed per condition, and these were used for both density and tensile measurements. The printing duration per sample was 29 min. The printed forms are shown in Figure 2. The overall density of all samples was measured based on the weight and dimensions measurements, and the water uptake at the end prior to tensile testing was lower than 5 wt.%, which is generally reported as being the usual amount observed in plant fibres [16].

The thermal behaviour of wood-based filament during the laying down process is captured using an infra-red camera (Flir A35 series from Flir company, Wilsonville, OH, USA). The experiment required the design of a particular sample geometry allowing for the analysis of thermal cycling without being penalised by the motion of the nozzle in front of the camera (Figure 2b). IR measurements were performed with a resolution of 320 × 256 pixels and a frame rate of 60 frames per second. Calibration of the IR camera was conducted by a contact thermocouple. The temperature mismatch at 190 °C was below 3 °C.

3D imaging of the microstructure of printed wood-based specimens was performed using X-ray micro-tomography. The equipment used to acquire the 3D microstructures was an UltraTom X-ray micro-CT from Rx-Solutions (Chavanod, France) using the following acquisition parameters: 230 KV X-ray source, 80 KV voltage, 480 µA current intensity, 1920 × 1536 pixel detector resolution, 1440 radiographic images, voxel size of 14.31 µm. The 3D microstructure was built using X-Act software from Rx-Solutions with a typical tomogram volume of 18 × 13 × 4 mm^3^ corresponding to 1277 × 906 × 291 voxels. All image processing protocols were performed using ImageJ software from NIH.

Tensile testing of 3D-printed wood-based specimens was performed using the same setup for the filament testing. As-printed specimens were tested to determine the effect of the printing temperature on Young’s modulus, tensile strength, and elongation at break. For the fracture toughness, notches were placed in tensile specimens using a circular saw with a typical depth of between 0.63 mm and 2.42 mm, representing 8.5% to 22.9% of the specimen’s width. The deformation sequences of all tested samples were recorded using a high-speed camera (Phantom V7.3 from Photonline, Marly Le Roi, 78-France). The typical frame rates used varied between 30 fps and 111,000 fps at full (800 × 600 pixels) and low (128 × 80 pixels) resolutions. 

Finally, the fractured samples were analysed using Scanning Electron microscopy (JEOL JSM 7600F microscope, Jeol Ltd, Akishima, Japan) under different magnifications from ×35 up to ×600 (pixel size between 0.15 µm and 2.60 µm).

## 3. Modelling Technique

The geometry considered for finite element computations was derived from 3D image acquisitions. Each solid voxel belonging to the solid phase was converted into a cuboid element with eight nodes per element. Each node has three degrees of freedom (dof), corresponding to its displacement components in X, Y and Z directions. Several FE models were built to study the effect of image resolution on the quality of the predictions. To adjust the resolution, the voxel size is increased from 20.44 µm to 143 µm. This corresponds to FE models to be solved with 0.85 × 10^6^ dof up to 172 × 10^6^ dofs. The FE model was considered to predict the elasticity behaviour of 3D printed wood-based specimens by implementing a linear elasticity model that was fully defined by the experimental Young’s modulus of the filament and its Poisson’s coefficient measured according to the image analysis of deformed samples. The use of these two engineering quantities means that there is an implicit assumption of isotropy for the filament properties. A more precise implementation of wood-based filament properties requires the transverse Young’s modulus value of the filament, together with the toolpath trajectory or the implementation of the wood properties within the filament itself, through more contrasted X-ray micro-tomography images. Although these two alternatives lead to a correction in the wood-based filament properties, they are out of reach in the context of the present study.

The boundary conditions refer to tensile loading conditions. These correspond to a fully constrained face against displacement in all directions and an extension in the longitudinal direction by a fixed amount (1% of the total length). The other degrees of freedom belonging to the loaded face are grounded. A Preconditioned Conjugate Gradient (PCG) solver was used to obtain the effective Young’s modulus ($E_{f}$) and stress intensity ($\sigma_{I}$). The computation cost for the quasi-static simulations varied between 1 and 229 min. The FE simulations were performed using Ansys software on a workstation equipped with 2 Xeon CPU operated at 3.0 Ghz and 1 Tbytes of RAM.

## 4. Results and Discussion

Figure 3 shows the thermal signature of the wood-based filament during the laying down process as a function of the printing temperature. The IR measurements reveal the rapid decrease of the filament temperature from the levels experienced at the nozzle exit to the ground values at the platform. Despite the rapid cooling of the filament, an improvement of the filament temperature during the laying down process is witnessed when the printing temperature is increased from 210 °C up to 240 °C. 

Figure 4 shows the thermal cycling captured at a fixed point situated at mid-length from the IR image. The evolution of the filament temperature is compared with the minimum and maximum printing temperatures. The thermal cycling reveals the main characteristics of the kinematics imposed on the printing nozzle. The first 150 s of the recording are typical cycles related to the production of the raft. These thermal cycles are distinguishable by having a larger period compared to the rest of the spectrum. The analysis of the remaining thermal cycles shows only a slight difference between the ground temperatures for the considered printing temperatures. The ground temperature is obtained when the sensed point is at the largest distance from the printing nozzle. The average ground temperature for 210 °C is only 41 °C. It increases to 46 °C when the printing temperature is moved to 240 °C, which means an improvement of only 12%. However, the peak filament temperature, which is obtained when the printing nozzle is coincident with the sensed point, reaches 88 °C for the lowest printing temperature. The same peak temperature increases to 153 °C for the highest printing temperature, which represents a substantial improvement (74%). Figure 4b compiles the average peak and ground temperatures for all printing conditions. Up to 220 °C, the difference between the peak and ground levels is nearly constant. The heat accumulation becomes more substantial for higher printing temperatures of up to 240 °C.

The overall density of the 3D-printed dog bone specimens are summarised in Table 2 as a function of the printing temperature. These results are obtained by weight and volume measurement. The density of the as-received filament is relatively low compared to PLA density (around 1.2 g/cm^3^) because of the presence of porosity revealed by SEM. The significance of this porosity is discussed by Le Duigo et al. [16] and further detailed later in this study. The overall density of the printed wood-based material varies between 0.90 g/cm^3^ and 0.95 g/cm^3^, depending on the printing temperature. This represents a decrease in the density of the printed specimens of 8% to 13% with respect to the density of the as-received filament. The accuracy of the density results according to the standard deviation over average criterion is 1.49%. This score demonstrates the reliability of the density measurement. In addition to the density measurements, sample volume mismatch with respect to the CAD model is summarised in Table 2. The positive values (from 5% to 7%) observed for the volume mismatch can be explained by the combination of dimensional variability induced by the processing and additional water uptake. The fact that this volume mismatch is positively correlated with the density means that swelling is more likely to be the major reason for obtaining larger samples. The variability of the volume mismatch results can be regarded as fairly acceptable based on the average standard deviation score of 16% with respect to the average values depicted for all printing temperatures. 

Figure 5 compares the tensile response of both as-received wood-based filament and 3D printed specimens under different printing conditions. The printed samples are not subject to notching prior to testing. The filament response is typical of a woody structure. The elasticity stage is marked by a series of damage events represented by sudden changes in the measured force.

This behaviour can be interpreted as intrinsic failure of the wood particles or interfacial decohesion between the PLA/PHA matrix and the wood reinforcement. This last hypothesis is more realistic with regard to the fracture patterns revealed by SEM of the printed structures. The wood filament additionally exhibits a small plasticity stage related to the contribution of PLA/PHA matrix.

A quasi-brittle failure is also a common point between all specimens. Finally, a large dispersion is observed between the filament responses due to the significant weight of the surface flaws in the tensile behaviour. The comparison between the wood-based filament with the pure PLA/PHA filament can be considered based on the tensile results for PLA/PHA published elsewhere [19]. The pure PLA/PHA filament used as a control exhibits high stretching capabilities compared to the woody filament, and the overall tensile response does not reveal any sudden changes in the measured force [19]. This confirms that the jagged behaviour of the wood-based filament is inherent to the wood filler behaviour. In addition, PLA/PHA filament also reaches larger tensile modulus and strength as shown in Table 2. Measured properties such as Young’s modulus, tensile strength and elongation at break are summarised in Table 2. Surprisingly, the dispersion of the results for the elongation at break is limited (2%) compared to Young’s modulus (14%) and tensile strength (21%). Young’s modulus of the tested wood filament (0.77 GPa) is significantly lower than that reported by Kariz et al. [14] (between 3 GPa and 3.27 GPa). The tensile strength results are larger in this study (32 MPa) compared to 30 MPa with a larger wood content (50%).

The tensile response of 3D-printed wood-based specimens shows a limited improvement of their performance when the printing temperature is increased (Figure 5b). The low tensile strength observed with nearly the same elongation at break is the main difference between the as-received filament and the printed one. This clearly indicates a loss in both stiffness and strength, even at the highest printing temperature. A secondary difference is the absence of sudden changes in force during the loading, suggesting a different deformation mechanism for the printed wood-based filament due to its higher level of structuring. Table 2 summarises the main extracted mechanical parameters for the printed specimens as a function of the printing temperature. The results in Table 2 suggest no clear trend with the printing temperature, with the exception that 220 °C and 230 °C seem to be optimal printing temperatures. The decrease of the tensile performance at high printing temperature can be explained by the defects that are triggered by the filament accumulation at the nozzle exit (Figure 3). According to Yang [15], thermal degradation of wood particles can be also considered a possible explanation for the decrease of the tensile performance of wood-based samples at high printing temperatures. In fact, the processing of the wood-based filament in an environment with high temperature (>210 °C) can be correlated with the decomposition of hemicellulose and cellulose and the partial decomposition of lignin that occurs within the range 210–370 °C. Even though the highest printing temperature considered in this study is below 260 °C, partial degradation of the wood particles is expected to affect the properties of FDM-printed samples. This degradation can be considered to be more pronounced for high printing temperatures according to the results of thermal cycling during the laying down process (Figure 3). The overall trend suggests a larger heat accumulation for the extruded filaments at temperatures above 220 °C. This may induce more significant degradation of the wood particles.

The loss in performance can be quantified as follows for the engineering constants. For the worst performing printing conditions, this loss represents 46%, 40% and 19% for Young’s modulus, tensile strength and elongation at break, respectively. For the best performing printing conditions, this loss decreases to 41%, 35% and 0% for the same engineering constants.

Figure 6a shows snapshots of the deformation sequences at the fracture point for 3D printed wood-based specimens as a function of the printing temperature. These specimens are pre-notched prior to testing. These sequences highlight a weak correlation between the raster and the fracture path. Indeed, only a jagged rupture profile is observed, which may indicate a limited contribution of the local shearing. The filament arrangement in the +45°/−45° sequence in the plane facing the camera (the plane of construction) is likely to trigger a strong dependence of the crack path on the filament orientation. However, this cannot be observed under all tested conditions. Figure 5b confirms the quasi-brittle trend and a limited effect of the printing temperature on the maximum stress if the printing condition 210 °C is excluded. Prior to failure, some signs of damage accumulations can be observed from the deformation sequences. The appearance of such damage and its development is behind the offset between the load point at maximum stress and the rupture point (Figure 6b). The fracture toughness derived from the tensile response in Figure 6b shows that this property improves with the increase of the printing temperature from 1 MPa × m1/2 up to 1.6 MPa × m1/2, which represents an improvement of 47%.

To measure the potential of wood-based filament with respect to the other polymers, data reported in [11,12,20,21,22,23,24,25,26,27,28,29,30,31,32,33,34] are compared to the properties summarised in Table 2 as a function of the printing temperature (Figure 7). It has to be mentioned that the wide apparent distribution revealed in Figure 7, especially for samples printed by thermoplastic materials without nature fibres (e.g., PLA, PP, ABS, PC, etc.) can be misleading. Indeed, a rough explanation would incorrectly suggest that the properties of FDM-printed materials are highly varied. In the present study, the results shown in Table 2 demonstrate that this is not the case. In fact, Figure 7 mostly reflects the variability expected from the use of different processing conditions in each study. Even if the printing temperature is regarded as an influential factor, the values of the other processing parameters vary from one study to another. The use of different printing speeds, filament segments with different layer heights, raw materials with different molecular weights, or samples with different sequences explains the wide distributions observed for each property investigated in Figure 7.

If the wide distribution of stiffness results reported for ABS and PLA is not considered, the wood-based filament has a low stiffness ranking within the limits of the dispersion of the experimental results (6% on average for Young’s modulus from the data in Table 2). It has a good ranking compared with the performance of ASA (Acrylonitrile styrene acrylate), PETG (Polyethylene terephthalate), Nylon, PLA/PHA, copolyester, etc. (Figure 7a).

However, the tensile strength of wood-based filament is limited compared to the polymers mentioned above (Figure 7b). This limited performance can be interpreted as a result of the high load of wood particles within the PLA matrix, which has been proved to significantly affect tensile strength, as shown by Kariz et al. [14]. Also, despite the good correlation between the printing temperature and the fracture toughness of printed wood-based specimens, this material ranks among the lowest-performing materials (Figure 7c). The main potential explanation for this ranking is the limited performance of the filament itself, which is transparent in terms of its tendency to induce damageable behaviour (Figure 5a). 

The SEM micrographs in Figure 8 provide some clues about the damageable behaviour of wood-based filament. Figure 8a shows the presence of micro-sized porosities within the filaments themselves with a typical size of about 74 ± 28 µm. This porosity is not generated by the printing process, but is rather inherent to the filament as suggested by Le Duigo et al. [16]. The initial porosity content reported by Le Duigo et al. [16] is 16.5%. In the present case, this porosity represents 12 ± 3% of the filament volume (Table 2). One possible reason explaining the presence of this porosity is the lack of a mixing state during the fabrication of the wood-based filament. This could be related, for instance, to the void left by a wood particle or a bubble trapped inside the matrix. Figure 8b–c shows magnified views of this genuine porosity of a nearly globular form. This porosity exhibits limited connectivity, and its rounded aspect with no cracks at the vicinity suggests an undamaged state. It is thus unlikely that cracks can be initiated from this spot although they surely contribute to the failure of the printed material. In addition to these genuine porosities, Figure 8d shows another type of porosity generated by the process. This represents gaps left by the laying down process and can be interpreted as a result of the mismatch between the adjacent filaments. Their characteristics are irregular, and they are larger compared to genuine porosities, with typical sizes between 170 µm and 270 µm. A less magnified view near the surface of the sample (Figure 8e) demonstrates greater connectivity between the process-induced porosities, especially close to the external frame. This connectivity remains below the millimetre scale (less than 300 µm). If both types of porosities are accounted for, the porosity content does not exceed 6% based on direct measurement on SEM micrographs. This amount is smaller compared to the levels between 15% and 22% reported by Le Duigo et al. [16]. The porosity levels for all printing conditions are given in Table 2. This porosity, which is derived from density measurements, only varies between 7% and 13%. It has no clear correlation with the printing conditions although the denser specimens seem to be the ones printed at high temperatures (>220 °C). 

Within the same micrograph, the aspect ratio between the lateral dimensions of the filament is 0.51 ± 0.05, which corresponds fairly well to the ratio between the filament diameter and the imposed layer height. The remaining part of the printed material is dense and presents signs of a quasi-brittle fracture, as attested by the limited tearing on the fractured surface in Figure 8f,g. The effect of tearing forces is limited to the interfacial region between PLA/PHA matrix and wood element, as illustrated by the interfacial decohesion in Figure 8h. From this SEM image, the wood particles appear as fibrous with a typical size between 30 and 60 µm. Although the particle size and distribution are known to affect the properties of composite, more quantitative information about the wood particle morphology and size distribution is, unfortunately, not available from the supplier’s datasheet of the as-received material.

To quantify the extent and distribution of the porosity within the printed wood-based specimens, X-ray micro-tomography results are discussed, below. Figure 9a shows cross-section views in all orthogonal directions. The dark grey levels are associated with the external air and the porosities within the printed specimen. Figure 9a confirms the presence of two types of porosity: that generated by the process and that within the filaments themselves. Within the plane of construction (XY), the porosity follows the raster orientation with a regular positioning along the +45° and –45° directions. 

From this image, there is no evidence of porosity localisation between the raster and the external frame. In the XZ image, the average roughness is 153 µm, which is of the same order of magnitude as the layer height (0.2 mm). This roughness is the main characteristic of the surface texture of the top and bottom faces of the printed material. Also, from this perspective, the pore connectivity seems to be limited in contrast with the YZ view (Figure 9a).

There is evidence of larger pore connectivity in the building direction due to the fact that there is a limited overlapping in the Z-direction of the void created within the raster. This means that the porosity connectivity is only driven by the process-induced porosity. The 3D view in Figure 9b shows the layered structure of the wood-based printed specimen and the porosity localisation within the raster. The image processing conducted to isolate the printed sample from the external air results in a more precise evaluation of the porosity content. For the printing temperature of 230 °C, the total amount of porosity is 3.92%. The amount measured in this study is lower than the amount measured from SEM, and it is at least four times lower than the result reported by Le Duigo et al. [16]. Sources of mismatch may be the low printing temperature of 210 °C considered in the cited study, or differences in terms of flow rate and layer thickness. Also, some genuine porosity identified from SEM micrographs has a typical size of about 36 µm, which is only 3 times the voxel size considered in this study. The pore connectivity quantified from the comparison between the volume of the largest pore and the total volume of the entire pore population shows that the connectivity measured is a short-range connectivity of about 0.1%. The size of the porosity ranges from the voxel size (14 µm) up to 270 µm. 

Figure 10 depicts the porosity profile along the main directions of the printed wood-based specimen. The jagged profiles in all directions indicate a certain periodicity of the porosity distribution due to the repeated filament trajectory. There is no significant anisotropy in the porosity distribution along the length of the sample although a varied content between 3% and 5% is observed. The same comment holds for the profile along the depth of the specimen. The porosity profile in the width direction presents two noticeable peaks culminating at 6% and 7%. These peaks are related to the surface porosity, which is partly built by the roughness in the XZ plane (Figure 9a). If the external face of the printed sample is considered to be a flat surface, then part of this surface porosity includes the external air. Within the same plane (XZ), the porosity content at the centre of the specimen reaches levels as large as 5% and lower values of 3% elsewhere. 

To measure the effect of the defects on the tensile performance of the printed wood-based structures, finite element computations based on the printed microstructures are discussed. 

Figure 11a shows the predicted stress intensity σ_I_ counterplot as a function of the resolution. The low resolution corresponds to the largest voxel size of 143 µm. 

Microstructural details at this resolution are not well captured, especially the genuine porosity and the layered form of the specimen. As a consequence, the stress heterogeneity partly reveals the effect of the filament arrangement within the plane of construction. When the voxel size is decreased to 72 µm, more details are implemented. The stress intensity distribution is more contrasted, especially through the thickness. Furthermore, by decreasing the voxel size to 48 µm, more details are added, but the difference becomes less evident compared to 72 µm. Three main features can be deduced from the analysis of this stress distribution. The first feature is the major role of the external frame, which appears as a load bearing element because of its denser features compared to the remaining part of the printed specimen. The second feature is the alternation of low and high stress levels within the raster, which indicates a strong correlation between the filament arrangement and the stress heterogeneity. Stress levels higher than 10 MPa can be found within the raster which is higher than the average levels within the external frame. The third feature is the limited stress localisation between the layers because the load direction is aligned with the length of the specimen.

Figure 11b depicts the evolution of the effective Young’s modulus as a function of the voxel size. In the same figure, the computation time was added to highlight the resource cost related to the FE computations based on real 3D microstructures. If the computations performed at the highest possible resolution (24 µm) are taken as a point of reference, then the predicted Young’s modulus deviation is only 2%. This means that the predictions are stable even at low resolutions. The reason for this stability is related to the small porosity content, which does not vary much by the lowering of the resolution. When the resolution is changed to the lowest one (voxel size of 143 µm), the average porosity content is 3.99%, which is 0.06% larger than the porosity content at the original resolution (voxel size of 14.31 µm). At the same time, the computation resources needed for fine resolutions increase exponentially. The comparison between the predicted value and the one measured for the same condition (printing temperature of 230 °C) shows that the predictions overestimate the stiffness of the printed wood-based material by nearly 27%. This overestimation could be related to the underestimation of the porosity effect, taking as a proof the larger porosity contents reported in. It could also be related to the material model implemented in FE computation, which stipulates a full load transfer between the layers and between the adjacent filaments.

To demonstrate the feasibility of printing a wood-based technical part on the centimetre scale (typical dimensions 45 × 48 × 2 mm^3^), the design shown in Figure 12a was considered.

The CAD model represents a 3D printing pen holder that fits the fixture of a Prusa i3 3D printer. The project aims to replace the original nozzle with a 3D printing pen of lower cost. The main steps up until the slicing stage are also shown in Figure 12a. The part orientation is selected to allow the production of the lowest amount of support material. The printing process was conducted using the optimal printing temperature (230 °C) according to the data shown in Table 2. The infra-red recording of the entire printing sequence (154 min) reveals a jagged raft production and a significant heat accumulation during the laying down of the wood-based filament (Figure 12b). The heat accumulation enhances the sticking of the part on the printing platform and reduces the risk of warping. This heat accumulation becomes more significant at the end of the printing sequence due to the limited length of the nozzle trajectory as can be seen from 126 min in Figure 12b. Despite the optimality of the part orientation, the support removal is tedious, and several imperfections can be noticed. These are mainly related to the stringing that is caused by a limited retraction of the filament. In addition, residual support is observed, especially during the shaping of the holes (Figure 12c). Despite these imperfections, the adaption of the pen holder on the commercial printer is a successful attempt (Figure 12d). The designed holder meets the expected in-service performance as it operates as a technical part for printing features from a modified 3D printer.

## 5. Conclusions

Wood-based filament made of PLA/PHA matrix reinforced by wood particles is printable in a wide range of temperatures ranging from 210 °C to 250 °C. Only a limited improvement in the tensile performance is obtained when the printing temperature is increased from 210 °C to 230 °C. Higher printing temperatures above 230 °C are not suitable, as the tensile properties can be affected by the thermal degradation of wood particles that occurs between 210 °C and 370 °C. This statement concludes on a sound cost-benefit ratio for low printing temperatures such as 220 °C, which should be selected in order to print parts in FDM using this feedstock material. The examination of the possible transfer of mechanical performance of the as-received filament to the printed specimens shows that only the elongation at break is fully restored. Therefore, there is an evident loss in the mechanical performance, which is mainly justified by a decrease of 41% and 35% in stiffness and strength, respectively, using optimal printing conditions. 

This study also concludes that the fracture mechanisms are not fully correlated with the raster, although jagged crack paths are observed. This is particularly due to the quasi-brittle nature of the filament itself. With regard to the performance of other polymers generally used in FDM, the major weakness of the studied wood-based filament is its limited tensile strength performance, which does not exceed 21 MPa. From a microstructural viewpoint, the FDM processing of the wood-based filament results in a complex microstructure, which is characterised by the presence of two populations of porosities; one is intrinsic to the filament and the other is process-induced. This study also concludes that this porosity has a limited connectivity and a relatively low content (4% according to X-ray micro-tomography results). The finite element computation reveals that stress heterogeneity is fully correlated with the filament arrangement within the plane of construction, along with the major role of the external frame in granting optimal mechanical stability and strength. In addition, it was found that the FE predictions overestimate the stiffness of the printed material by 27%, suggesting that there should be greater room for the effect of porosity or the interfacial effect between the adjacent filaments. Finally, it can be concluded that wood-based filament should be targeted for the production of customised features exhibiting an atypical texture where the performance is not the first design criterion.

## Figures and Tables

**Figure 1 polymers-11-01778-f001:**
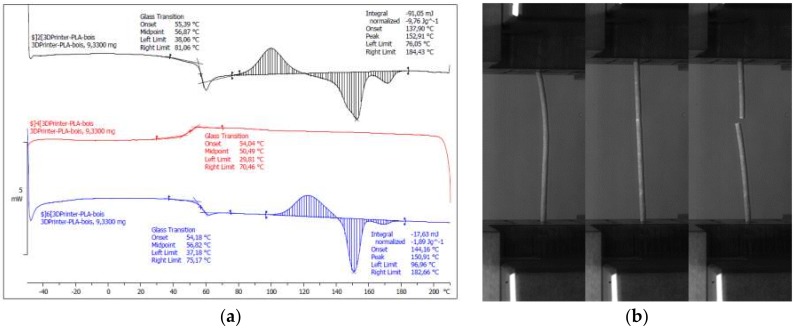
Thermal and mechanical properties of wood-PLA/PHA prior 3D printing. (**a**) Thermal analysis using DSC, (**b**) tensile testing of the filament.

**Figure 2 polymers-11-01778-f002:**
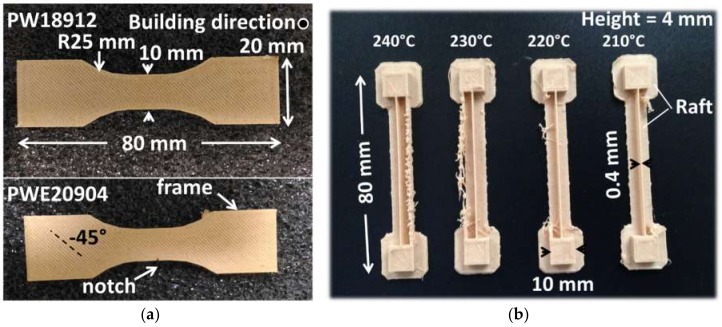
(**a**) Unnotched and notched tensile specimens printed using wood-PLA/PHA filament. Notches were performed after printing. (**b**) Specimen considered for thermal analysis under different printing temperatures.

**Figure 3 polymers-11-01778-f003:**
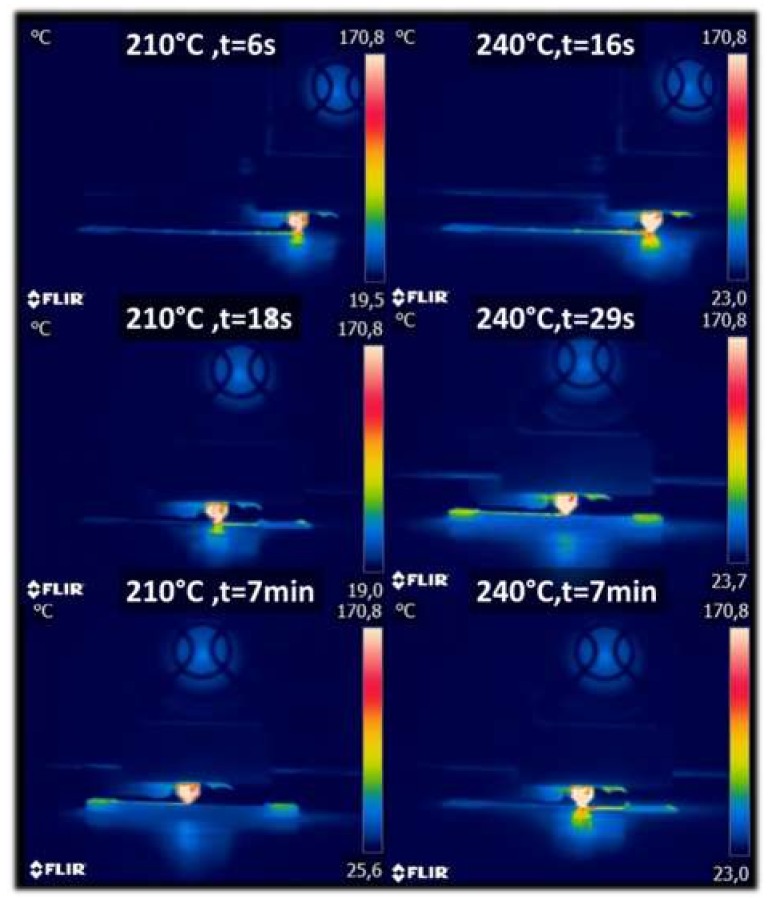
Laying down process of wood-PLA/PHA captured using infra-red camera for two distinct printing temperatures.

**Figure 4 polymers-11-01778-f004:**
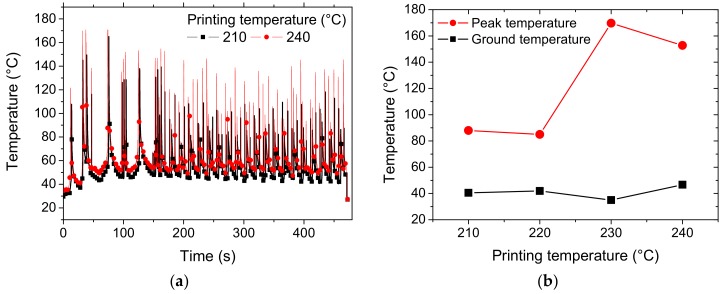
Results of infra-red measurement of the laying down process of wood-PLA/PHA filament: (**a**) thermal cycles for the minimum and maximum printing temperatures, (**b**) extracted ground and peak temperatures for all attempted printing temperatures.

**Figure 5 polymers-11-01778-f005:**
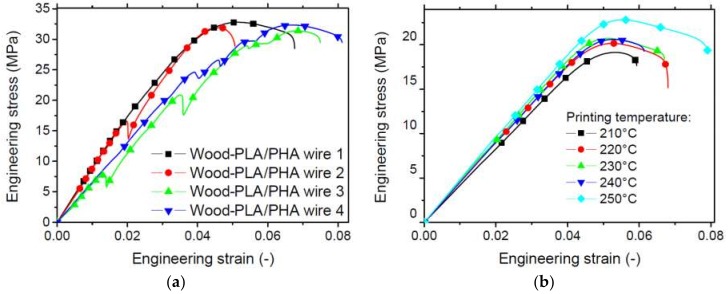
Comparison between the tensile response of (**a**) as-received wood-PLA/PHA filament and (**b**) 3D-printed specimens of the same material.

**Figure 6 polymers-11-01778-f006:**
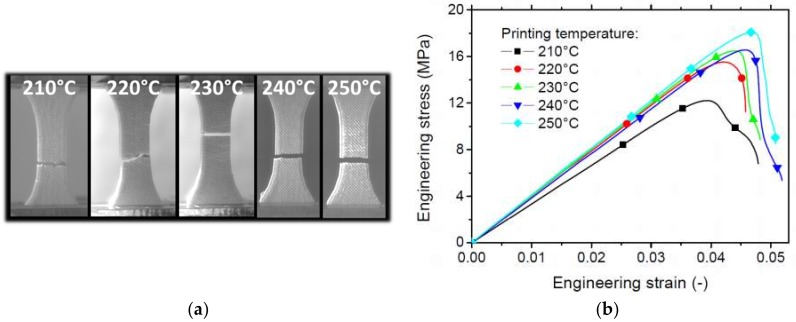
(**a**) Crack patterns at the rupture point of notched wood-PLA/PHA as a function of the printing temperature, and (**b**) corresponding tensile response.

**Figure 7 polymers-11-01778-f007:**
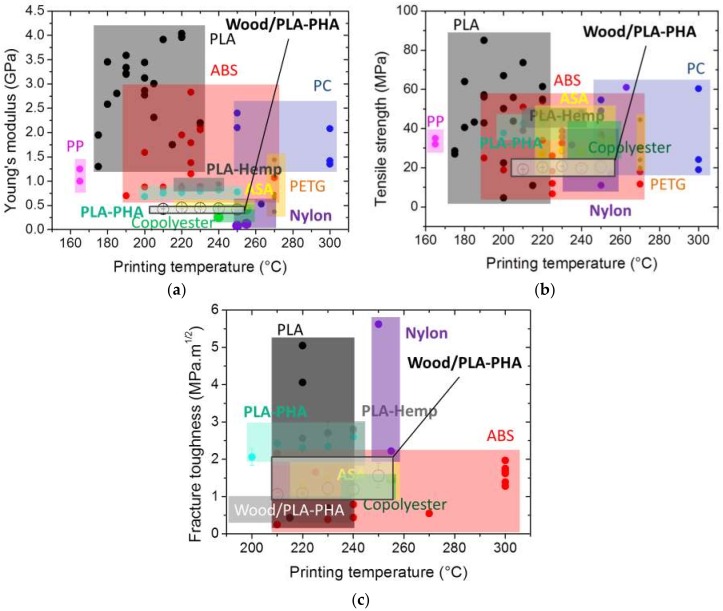
Tensile performance of varieties of polymers used in FDM including the wood-PLA/PHA considered in this study. (**a**) Stiffness, (**b**) strength, and (**c**) toughness. The performance of wood-based filament is framed.

**Figure 8 polymers-11-01778-f008:**
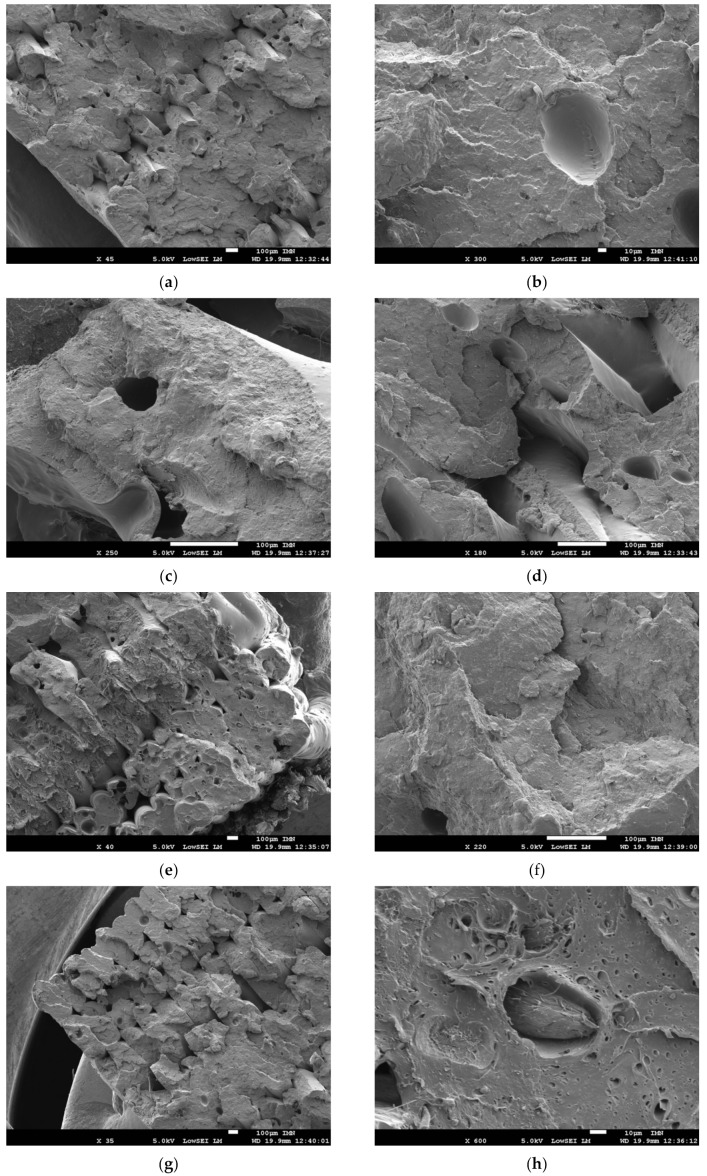
SEM micrographs of the fractured surfaces in 3D printed Wood-PLA/PHA after tensile loading showing different features: (**a**) raster, (**b**) process-induced porosity, (**c**) frame, (**d**) wood fibre, (**e**) fibre pull-out, (**f**) PLA/PHA matrix, (**g**) filament morphology, (**h**) fracture pattern within the matrix. The printing temperature is 250 °C.

**Figure 9 polymers-11-01778-f009:**
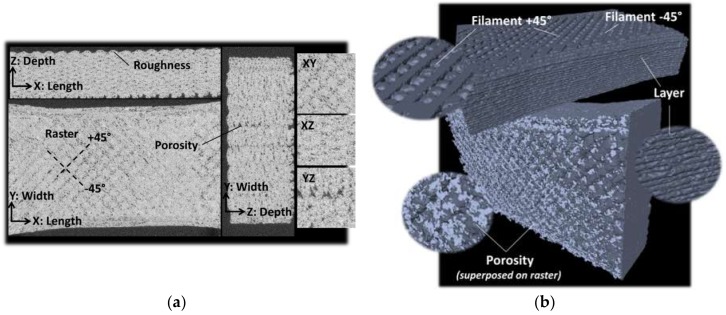
Illustration of X-ray micro-tomography results: (**a**) cross-section views in the three orthogonal planes showing the main features in 3D printed wood-PLA/PHA, (**b**) 3D view of the layer build-up showing the surface texture and the network of process-induced porosity.

**Figure 10 polymers-11-01778-f010:**
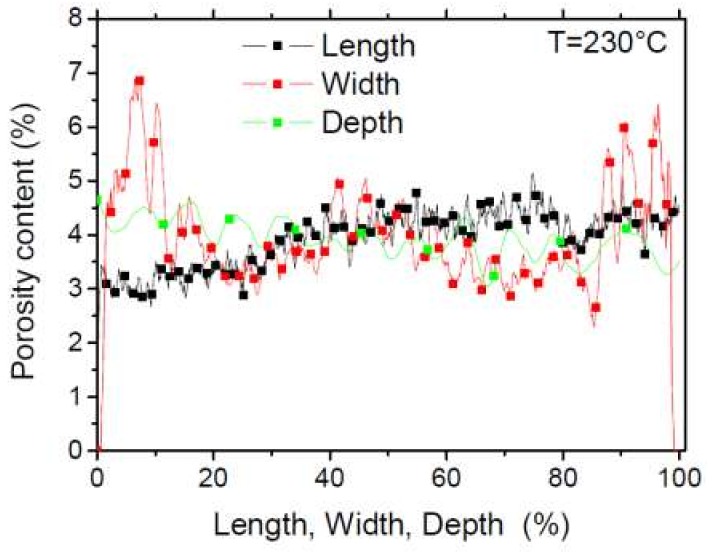
Porosity content in the three main directions in printed wood-PLA/PHA.

**Figure 11 polymers-11-01778-f011:**
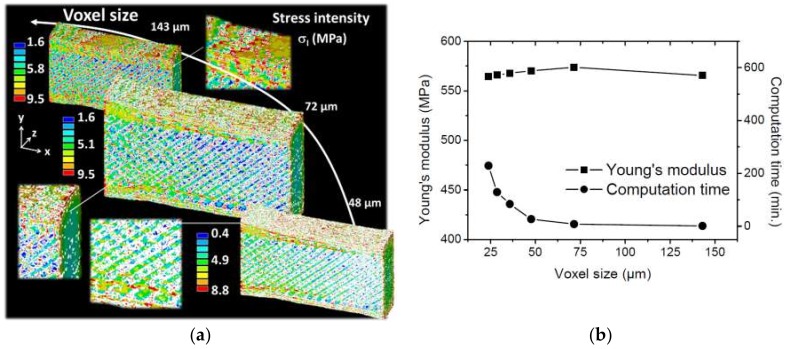
(**a**) Prediction of the stress intensity distribution in 3D printed Wood-PLA/PHA using finite element computation as a function of the voxel size. (**b**) Influence of the resolution on the predicted Young’s modulus of wood-PLA/PHA printed at 230 °C. The computation time trend is also shown.

**Figure 12 polymers-11-01778-f012:**
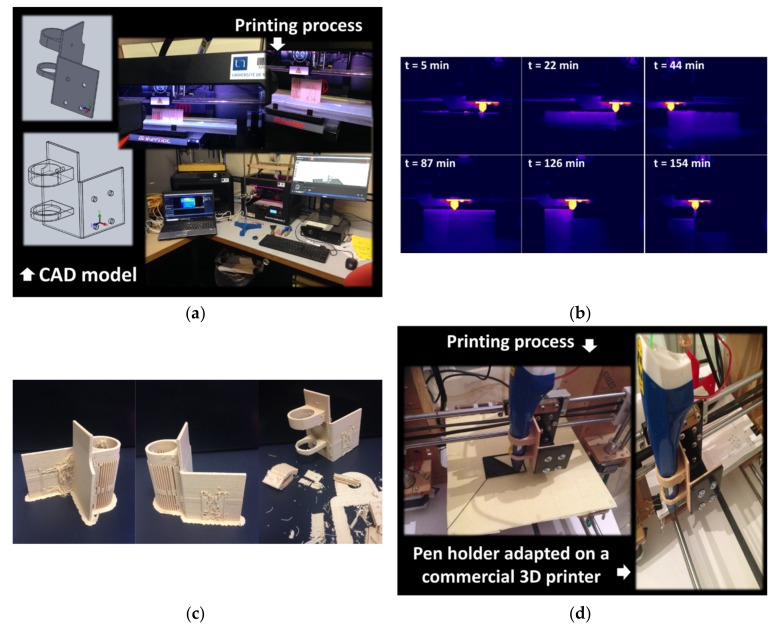
(**a**) Mains steps for designing a 3D printing pen holder on a commercial printer, (**b**) infra-red monitoring during the laying down process, (**c**) final rendering of the part prior to and after support removal, (**d**) part fixation and performance during service.

**Table 1 polymers-11-01778-t001:** Printing parameters used in this study.

Support		Printing		Extruder		Raft	
density	0.2	extruder temperature	varied	filament diameter	1.75 mm	raft to model spacing	0.35 mm
margin	0.5 mm	bed temperature	25°	retraction distance	1.3 mm	raft margin	4 mm
support to model spacing	0.4 mm	travel speed	150 mm/s	retraction speed	25 mm/s	base pattern spacing	0.8 mm
support angle	68°	Z-axis travel speed	23 mm/s	infill density	100%	base pattern length	15 mm
support layer thickness	0.2 mm	Sample orientation	building // thickness	nozzle diameter	0.4 mm	base layer density	0.7
		layer height	0.2 mm	layup	+45°/−45°	base layer height	0.3 mm

**Table 2 polymers-11-01778-t002:** Tensile properties of as-received and printed Wood-PLA/PHA.

Material	Printing Temperature °C	Density (G/Cm^3^)	Volume Mismatch (%)	Porosity Content ^1^ (%)	Young’s Modulus (Mpa)	Tensile Strength (Mpa)	Elongation at Break (%)	Fracture Toughness Mpa × M1/2
PLA/PHA	-	1.11 ± 0.00	-	0	1034±132	46 ± 2	71 ± 6	-
as-received	-	1.03	-	12 ± 3.0	771 ± 110	32.1 ± 0.6	6.9 ± 1.30	-
printed	210	0.90 ± 0.00	4.93±0.29	13 ± 1.6	426 ± 11	19.2 ± 0.1	6.03 ± 0.14	1.06 ± 0.05
	220	0.92 ± 0.00	5.02±1.11	10 ± 1.1	453 ± 19	20.3 ± 0.1	6.29 ± 0.72	1.08 ± 0.01
	230	0.93 ± 0.02	5.76±1.95	9 ± 1.9	446 ± 20	20.8 ± 0.1	6.70 ± 0.01	1.22 ± 0.17
	240	0.92 ± 0.01	6.79±1.17	11 ± 2.2	438 ± 10	19.8 ± 1.2	6.15 ± 0.03	1.19 ± 0.11
	250	0.95 ± 0.04	^2^ -	7 ± 3.1	416 ±76	20.5 ± 3.2	7.06 ± 1.21	1.56 ± 0.33

^1^ density and SEM measurements, ^2^ unsuccessful.

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
