# Peer review of "Microstructure and Mechanical Performance of 3D Printed Wood-PLA/PHA Using Fused Deposition Modelling: Effect of Printing Temperature"

_polymers, 2019, doi:10.3390/polym11111778_

Round 1

Reviewer 1 Report

The paper presents an interesting research and shows new informations about 3d printed materials. This findings are valuable for people dealing with 3d printing and modelling the products made by 3d printing.

Maybe the printing temperature should be incorporated in the title of the paper, since it was its effect on 3d printed parts properties researched in the research?

It would be good to have a control, for example pure PLA/PHA filament without wood to compare the results. Probably this is a part of different research?

I am also missing the number of the specimens measured/tested for each data (number of specimens for measuring density or porosity…etc).

Some comments:

Line 50: … between the filaments [8]…

Maybe find the better word for filaments? Since filaments are  before nozzle, but after extrusion through nozzle is not a filament anymore…

Line 66: …. Kariza et al. [13] formulated…

Comment: Misspelling author surname-without a on the end: Kariz et al.

Line 79:  The feedstock material is a 30% recycled pinewood fibre - 70% PLA/PHA

Comment: Is there any information about particle size and distribution? Fibre or powder- this really effects the properties of composite. Since it is a commercial material - are there any informations about other additives in the filament?

Line 158: … The FE simulation are performed on a workstation equipped with 2 Xeon CPU operated at 3.0 Ghz and 1 Tbytes of RAM…

Comment: Which software was used for simulations?

Line 196: … varies between 0.90 g/cm3 and 0.95 g/cm3 and 1.05 g/cm3…

Comment: if I am looking at table 2, I do not see density 1,05 g/cm3?

Author Response

Reviewer 1 :

The paper presents an interesting research and shows new informations about 3d printed materials. This findings are valuable for people dealing with 3d printing and modelling the products made by 3d printing.

We thank the reviewer for his positive opinion about our work.

Maybe the printing temperature should be incorporated in the title of the paper, since it was its effect on 3d printed parts properties researched in the research?

We added the printing temperature in the title as requested by the reviewer

It would be good to have a control, for example pure PLA/PHA filament without wood to compare the results. Probably this is a part of different research?

This is a good idea. We added the PLA/PHA tensile properties based on the results of a former work. In addition, we discussed the differences between the reference material and the Wood-based filament.

Amendment in section 4: “The comparison between the wood-based filament with the pure PLA/PHA filament can be considered based on the tensile results for PLA/PHA published elsewhere (17). The pure PLA/PHA filament taken as a control exhibits large stretching capabilities compared to the woody filament and the overall tensile response does not reveal any sudden changes in the reaction force (17). This confirms that the jagged behavior of the wood-based filament is inherent to the wood filler behavior. In addition, PLA-PHA filament also reaches larger tensile modulus and strength as shown in Table 2.”

+ modified table 2 to consider the data for PLA-PHA filament

+ added reference: Guessasma S, Belhabib S, and Nouri H. Thermal cycling, microstructure and tensile performance of PLA-PHA polymer printed using fused deposition modelling technique, Rapid Prototyping Journal 2019;in press.

I am also missing the number of the specimens measured/tested for each data (number of specimens for measuring density or porosity…etc).

Four samples were printed per condition. These were used for both density and tensile testing experiments.

Amendment in section 1: “Four samples are printed per condition and these are used for both density and tensile measurements.”

Some comments:

Line 50: … between the filaments [8]…

Corrected. The new sentence looks like “: the stress concentration that develops around the process-generated porosities, and the filament decohesion that occurs at the interface”

Maybe find the better word for filaments? Since filaments are  before nozzle, but after extrusion through nozzle is not a filament anymore…

 The term filament is commonly admitted in the literature of additive manufacturing even after the laying down. If we use another term this will be misleading. In fact, the arrangement of the filaments is still valid after processing since FDM does not allow continuity of the matter and filament morphology can be even captured. In order to differentiate between the two states, the as-received or raw filament is used for the wire prior processing and the term filament or extruded filament is used for the processed one. Corrections were made thought out the text in multiple positions.

Line 66: …. Kariza et al. [13] formulated…

Comment: Misspelling author surname-without a on the end: Kariz et al.

corrected

Line 79:  The feedstock material is a 30% recycled pinewood fibre - 70% PLA/PHA

Comment: Is there any information about particle size and distribution? Fibre or powder- this really effects the properties of composite. Since it is a commercial material - are there any informations about other additives in the filament?

The reviewer is right about the effect of the particle size. Unfortunately, the supplier did not provide this information about the filler morphology. From our SEM images, we can see (Figure 8h) that the wood particles have a fibrous morphology with a typical size between 30 and 60 µm.

Amendment in section 4: “From this SEM image, the wood particles appear as fibrous with a typical size between 30 and 60 µm. Although, the particle size and distribution are known to affect the properties of composite, a more quantitative information about the wood particle morphology and size distribution is not available.”

Line 158: … The FE simulation are performed on a workstation equipped with 2 Xeon CPU operated at 3.0 Ghz and 1 Tbytes of RAM…

Comment: Which software was used for simulations?

Ansys finite element software was used for the simulations.

Amendment in section 3: “The FE simulations are performed using Ansys software on a workstation equipped with 2 Xeon CPU operated at 3.0 Ghz and 1 Tbytes of RAM.”

Line 196: … varies between 0.90 g/cm3 and 0.95 g/cm3 and 1.05 g/cm3…

Comment: if I am looking at table 2, I do not see density 1,05 g/cm3?

Sorry this was a mistake the right sentence is “The overall density of the printed wood-based material varies between 0.90 g/cm3 and 0.95 g/cm3 depending on the printing temperature. This represents from 8% to 13% of decrease in the density of the printed specimens with respect to the”. This now corrected.

Reviewer 2 Report

The manuscript ''Microstructure and mechanical performances of 3D printed wood/PLA/PHA using fused deposition modelling'' mainly described effects of the printing temperature on properties of 3D printed wood/PLA/PHA composites using FDM. To my knowledge, effects of the printing temperature on properties of FDM-printed wood plastic composites (WPCs) have rarely been discussed before. However, there are a few confusions in the article, and the careful discussion of the results are required. There are a few comments as follows:

Comment 1: The title of the article is suggested to be revised to emerge the significance of the study. The content in the article is most discussed about effects of the printing temperatures on performances of WPC. Therefore, it is recommended to insert words, such as temperature, in the title and the keywords as to show the true value of the original work.

Comment 2: There are two papers related to performances of FDM-printed WPC filaments, which are published in Polymers recently. It is recommended to cite these papers in the introduction. (1) Mazzanti, Valentina; Malagutti, Lorenzo; Mollica, Francesco. FDM 3D Printing of Polymers Containing Natural Fillers: A Review of their Mechanical Properties. Polymers 2019, 11, 1094. (2)Teng-Chun Yang. Effect of extrusion temperature on the physico-mechanical properties of unidirectional wood fiber-reinforced polylactic acid composite (WFRPC) components using fused deposition modeling. Polymers 2018, 10, 976.

Comment 3: It can be seen that the data with deviation in Table 2. Please describe how many numbers of samples measured in each test. Additionally, it should perform statistical analysis on data because some properties of the samples after the same testing may be sufficient or insufficient difference.

Comment 4: In Figure 7, it shows properties of FDM-printed WPC as a function of the printing temperature are compared to the other previous researches. It can be seen that the data is very dispersion in each printed sample, especially for samples printed by thermoplastic materials without nature fibers (ex. PLA, PP, ABS, PC etc.). It could be influenced by different conditions for FDM printing in each research, such as different printing speed, filament segment with different layer height, raw material with different molecular weight, samples with different sequence, etc. Therefore, this figure could mislead the readers to think that the properties of FDM-printed materials are highly variable. Please the authors explain in detail or revise the figure.

Comment 5: Most of the previous researches reported that the processing environment with high temperature (> 160oC) causes the degradation of wood. Therefore, the high printing temperature (> 200oC) could more or less cause the degradation of wood fibers to affect the properties of FDM-printed WPC samples. In the overall content, the authors did not mention about effects of the printing temperature on wood fibers. Please discuss about this issue.

Comment 6: In Figure 3, the thermal signature of the filament during the laying down process as a function of the printing temperature is shown. In the article, the authors did not explain the effect of these results on the properties of FDM-printed WPC samples. Please add to describe it. Otherwise, these results just implied the performance of the 3D printing machine.

Author Response

Reviewer 2 :

The manuscript ''Microstructure and mechanical performances of 3D printed wood/PLA/PHA using fused deposition modelling'' mainly described effects of the printing temperature on properties of 3D printed wood/PLA/PHA composites using FDM. To my knowledge, effects of the printing temperature on properties of FDM-printed wood plastic composites (WPCs) have rarely been discussed before. However, there are a few confusions in the article, and the careful discussion of the results are required. There are a few comments as follows:

We agree with the opinion of the reviewer, the effect of the printing temperature is more highlighted in this new version.

Comment 1: The title of the article is suggested to be revised to emerge the significance of the study. The content in the article is most discussed about effects of the printing temperatures on performances of WPC. Therefore, it is recommended to insert words, such as temperature, in the title and the keywords as to show the true value of the original work.

The title was changed according to the remark by the reviewer. Now the new title is: Microstructure and mechanical performance of 3D printed wood/PLA/PHA using fused deposition modelling: effect of printing temperature

We also added the printing temperature in the keyword list.

Comment 2: There are two papers related to performances of FDM-printed WPC filaments, which are published in Polymers recently. It is recommended to cite these papers in the introduction. (1) Mazzanti, Valentina; Malagutti, Lorenzo; Mollica, Francesco. FDM 3D Printing of Polymers Containing Natural Fillers: A Review of their Mechanical Properties. Polymers 2019, 11, 1094. (2)Teng-Chun Yang. Effect of extrusion temperature on the physico-mechanical properties of unidirectional wood fiber-reinforced polylactic acid composite (WFRPC) components using fused deposition modeling. Polymers 2018, 10, 976.

We thank the reviewer for these valuable references that we were not aware of. These citations are now considered in the introduction section.

Amandement in section 1: “The recent review by Mazzanti et al. [13] showed that the use of natural filler is a promising route for the design of environmentally-friendly components that exhibit superior performance. The analysis of the literature works in the same review paper demonstrated the lack of impacting research on biofillers combined with PLA or ABS compared to less known polymers such as polyolefins.” And “.Yang considered a wood fibre filler in PLA as a feedstock material in FDM [15]. The author showed that the mechanical properties were negatively affected by printing temperatures larger than 200°C while most physical properties including the density of the printed composite increased with the increase of the printing temperature in the range 200°C – 230°C.”

Added references:

Mazzanti V, Malagutti L, and Mollica F. Polymers 2019;11(7):1094.

Yang T-C. Polymers 2018;10(9):976.

Comment 3: It can be seen that the data with deviation in Table 2. Please describe how many numbers of samples measured in each test. Additionally, it should perform statistical analysis on data because some properties of the samples after the same testing may be sufficient or insufficient difference.

This comment is similar to the former comment from reviewer 1. We have used four samples as discussed earlier. However, we believe that the statistical analysis such as with ANOVA is only relevant if the number of influential factors is larger than 2. In our case, the only varied parameter is the printing temperature. The analysis of variability of the results was conducted in the new version based on the reviewer comment for the data given in Table 2.

Multiple amendments in section 4: “The accuracy of the density results according to the standard deviation over average criterion is 1.49%. This score demonstrates the reliability of the density measurement.” + “The variability of the volume mismatch results can be considered as fairly acceptable based on the average standard deviation score of 16% with respect to the average values depicted for all printing temperatures.” + (from the former version) “Surprisingly, the dispersion of the results for the elongation at break is limited (2%) compared to Young’s modulus (14%) and tensile strength (21%).”

Comment 4: In Figure 7, it shows properties of FDM-printed WPC as a function of the printing temperature are compared to the other previous researches. It can be seen that the data is very dispersion in each printed sample, especially for samples printed by thermoplastic materials without nature fibers (ex. PLA, PP, ABS, PC etc.). It could be influenced by different conditions for FDM printing in each research, such as different printing speed, filament segment with different layer height, raw material with different molecular weight, samples with different sequence, etc. Therefore, this figure could mislead the readers to think that the properties of FDM-printed materials are highly variable. Please the authors explain in detail or revise the figure.

The reviewer is right. The data provided concern various printing conditions which do not reflect he variability assessed from a particular combination of FDM processing parameters. In order to highlight more this aspect Figure 7 is kept and further explanation is provided.

Amendment in section 4: “It has to be mentioned that the apparent large dispersion revealed in Figure 7, especially for samples printed by thermoplastic materials without nature fibres (ex. PLA, PP, ABS, PC etc.) can be misleading. Indeed, a rough explanation would wrongly suggest that the properties of FDM-printed materials are highly varied. In the present study, results shown in Table demonstrate that this is not the case. In fact, Figure 7 reflect mostly the variability expected from the use of different processing conditions in each research. Even if the printing temperature is considered as an influential factor, the values of the other processing parameters vary from one research to another. The use of different printing speed, filament segment with different layer height, raw material with different molecular weight, or samples with different sequence explains the large bounds observed for each property investigated in Figure 7.”

Comment 5: Most of the previous researches reported that the processing environment with high temperature (> 160oC) causes the degradation of wood. Therefore, the high printing temperature (> 200oC) could more or less cause the degradation of wood fibers to affect the properties of FDM-printed WPC samples. In the overall content, the authors did not mention about effects of the printing temperature on wood fibers. Please discuss about this issue.

The reviewer is right. We did not mention the thermal degradation as a possible cause for the low performance of the wood-based structures at high printing temperatures because we were not sure. This is because we did not run advanced thermal analysis. The only thermal changes of the wood-based filaments that we were able to capture from our DSC results show that there is mainly changes related to water activity up to 200°C. From our mechanical testing results we observe that the optimal printing temperatures are between 220°C and 230°C (see table 2 and the discussion in section 4) not below 200°C. We observed that in the average printing temperatures larger than 230°C are not suitable. In the new version we take further steps and relate our findings to the possible thermal degradation of wood fibers based on the literature results especialy those provided by the reviewer (Yang Polymers 2018).

Amendment: in section 4 “According to Yang [15], thermal degradation of wood particles can be also considered as a possible explanation for the decrease of the tensile performance of wood-based samples at high printing temperature. In fact, the processing of the wood-based filament in an   environment with high temperature (> 210°C) can be correlated to the decomposition of hemicellulose and cellulose and the partial decomposition of lignin that occurs within the range 210°C – 370°C. Even if the largest printing temperature considered in this study is below 260°C, partial degradation of the wood particles is expected to affect the properties of FDM-printed samples.” + conclusion section “when the printing temperature is increased between 210°C and 230°C. Higher printing temperatures above 230°C are not suitable as the tensile properties are be affected by the thermal degradation of wood particles that occurs between 210°C and 370°C.”

Comment 6: In Figure 3, the thermal signature of the filament during the laying down process as a function of the printing temperature is shown. In the article, the authors did not explain the effect of these results on the properties of FDM-printed WPC samples. Please add to describe it. Otherwise, these results just implied the performance of the 3D printing machine.

The reviewer is right. We discussed the effect of heat accumulation on the possible filament degradation, especially for high temperatures.

Amendment in section 4: “This degradation can be considered as more pronounced for high printing temperatures according to the results of thermal cycling during the laying down process (Figure 3). The overall trend suggests a larger heat accumulation for those extruded filaments at temperatures above 220°C. This may induce a more significant degradation of the wood particles.”

Reviewer 3 Report

Author could improve the English and grammatical 

line no 109 check the word "Thanks"

Author could write boundary conditions in the Methodology section

Author may validate your results with published literature

Author Response

Reviewer 3 :

Author could improve the English and grammatical 

We checked again the manuscript against lingual errors. All corrections are highlighted in the new version.

line no 109 check the word "Thanks"

The full sentence was revised as follows “The thermal behaviour of wood-based filament during the laying down process is captured using an infra-red camera (Flir A35 series from Flir company).”

Author could write boundary conditions in the Methodology section

The boundary conditions of the finite element computations were already specified in the section 3. Check the following paragraph “The boundary conditions refer to tensile loading conditions. These correspond to a fully constrained face against displacement in all directions and an extension in the longitudinal direction by a fixed amount (1% of the total length). The other degrees of freedom belonging to the loaded face are grounded.”

Author may validate your results with published literature

We discussed our achievements based on several published results from the literature. The references used for such a discussion are given in the list : 11,12, 14, 16, 20-34.

Round 2

Reviewer 2 Report

I think that the authors have improved and revised their article based on my comments. I recommend accepting this article.